# Design and Evaluation of ^223^Ra-Labeled and Anti-PSMA Targeted NaA Nanozeolites for Prostate Cancer Therapy–Part I

**DOI:** 10.3390/ma13173875

**Published:** 2020-09-02

**Authors:** Malwina Czerwińska, Giulio Fracasso, Marek Pruszyński, Aleksander Bilewicz, Marcin Kruszewski, Agnieszka Majkowska-Pilip, Anna Lankoff

**Affiliations:** 1Centre for Radiobiology and Biological Dosimetry, Institute of Nuclear Chemistry and Technology, Dorodna 16, 03-195 Warsaw, Poland; m.wasilewska@ichtj.waw.pl (M.C.); m.kruszewski@ichtj.waw.pl (M.K.); 2Department of Medicine, University of Verona, Piazzale LA Scuro 10, 37134 Verona, Italy; giulio.fracasso@univr.it; 3Centre of Radiochemistry and Nuclear Chemistry, Institute of Nuclear Chemistry and Technology, Dorodna 16, 03-195 Warsaw, Poland; m.pruszynski@ichtj.waw.pl (M.P.); a.bilewicz@ichtj.waw.pl (A.B.); a.majkowska@ichtj.waw.pl (A.M.-P.); 4Faculty of Chemistry, University of Warsaw, Pasteura 1, 02-093 Warsaw, Poland; 5Department of Molecular Biology and Translational Research, Institute of Rural Health, Jaczewskiego 2, 20-090 Lublin, Poland; 6Department of Medical Biology, Institute of Biology, Jan Kochanowski University, Uniwersytecka 7, 24-406 Kielce, Poland

**Keywords:** nanozeolite NaA, radium-223, D2B anti-PSMA antibody, targeted prostate cancer therapy, in vitro studies

## Abstract

Prostate cancer is the second most frequent malignancy in men worldwide. Unfortunately, current therapies often lead to the onset of metastatic castration-resistant prostate cancer (mCRPC), causing significant mortality. Therefore, there is an urgent need for new and targeted therapies that are advantageous over the current ones. Recently, the PSMA-targeted radioligand therapy of mCRPC has shown very promising results. In line with this, we described the synthesis of a new radioimmunoconjugate, ^223^RaA-silane-PEG-D2B, for targeted mCRPC therapy. The new compound consists of a NaA zeolite nanocarrier loaded with the α-particle emitting Ra-223 radionuclide, functionalized with the anti-PSMA D2B antibody. Physicochemical properties of the synthesized compound were characterized by standard methods (HR-SEM, TEM, XRD, FTIR, EDS, NTA, DLS, BET, TGA). The targeting selectivity, the extent of internalization, and cytotoxicity were determined in LNCaP C4-2 (PSMA+) and DU-145 (PSMA-) cells. Our results supported the ^223^RaA-silane-PEG-D2B synthesis and revealed that the final product had a diameter ca. 120 nm and specific activity 0.65 MBq/1mg. The product was characterized by a high yield of stability (>95% up to 12 days). The conjugation reaction resulted in approximately 50 antibodies/nanoparticle. The obtained radioimmunoconjugate bound specifically and internalized into PSMA-expressing LNCaP C4-2 cells, but not into PSMA-negative DU-145 cells. ^223^RaA-silane-PEG-D2B demonstrated also potent cytotoxicity in LNCaP C4-2 cells. These promising results require further in vivo evaluation of ^223^RaA-silane-PEG-D2B with regard to its toxicity and therapeutic efficacy.

## 1. Introduction

Prostate cancer (PCa) is the second most frequent malignancy in men worldwide and the second leading cause of cancer-related deaths in Western civilization [1]. The standard treatment of this cancer is based on radical prostatectomy, external beam radiation therapy, or brachytherapy, chemotherapy, and hormonotherapy. Unfortunately, these therapies are often followed by the formation of metastatic castration-resistant prostate cancer (mCRPC) [2]. During the last few years, several options for the treatment of mCRPC have shown a survival benefit in clinical trials, e.g., CYP17 inhibitor-abiraterone, androgen signaling inhibitor-enzalutamide, taxane cabazitaxel, the bone-seeking radium-223 chloride, and vaccine - Sipuleucel-T [3]. Unfortunately, despite several attempts, the median survival for men with mCRPC ranges from 13 to 23 months [4]. Due to the significant mortality and morbidity rate associated with the progression of this disease, there is an urgent need for new and targeted treatments. So far, several cell-surface associated proteins, such as glycoproteins, receptors, enzymes, and peptides, have been investigated as targets for the treatment of prostate cancer [5]. One of these is the prostate-specific membrane antigen (PSMA). PSMA is a well-characterized target that is overexpressed selectively on prostate cancer cells and in the neovasculature of several types of solid tumors but has limited expression in normal non-prostatic tissues [6,7,8,9]. Its expression increases progressively with higher-grade prostate cancer, metastatic disease, and mCRPC [10]. PSMA seems also to drive oncogenic signaling in prostate cancer cells and, therefore, to have a role in PCa aggressiveness [11,12]. PSMA functions as a cell surface receptor with a large extracellular domain, which allows for effective antibody access. Moreover, this receptor efficiently internalizes the bound antibody and any payload attached to it [13,14]. These findings have spurred the development of many antibodies against PSMA. Of these, the mAbs J591, J533, D2B, E99, 3/A12, 3/E7, and 3/F11 have been shown to bind most efficiently to PSMA present on the cell surface [15] and have been used in preclinical and clinical studies [16]. These studies have shown encouraging results of radiotherapy with PSMA ligands labeled with beta emitters, including yttrium-90, lutetium-177, iodine-131, and terbium-161. The clinical trials included approximately 800 mCRPC patients treated with PSMA antibodies or PSMA small molecule inhibitors labeled mainly with yttrium-90 and lutetium-177 (^90^Y-huJ591, ^177^Lu-huJ591, ^177^Lu-PSMA-617, ^177^Lu-PSMA I&T). Though this type of therapy has allowed efficient treatment of large tumors, it has been suboptimal for the eradication of single metastatic cells or small cell clusters [17]. However, it has been found that up to 30% of patients have never responded to this therapy or developed resistance. Moreover, hematological toxicity has been reported as a prevalent serious side effect [18,19,20]. Recent studies have shown that therapy with PSMA ligands labeled with the alpha emitters can overcome resistance to therapy with beta emitters [21]. First of all, α particles are much more radiotoxic than β particles due to the higher value of linear energy transfer (LET) (50–230 keV/μm vs. 0.2–2 keV/μm). Secondly, the tissue range of α particles is much shorter than β particles (0.05–0.1 mm vs. 2–10 mm), what allows their application to small, disseminated tumors, micro-metastases, or individual, circulating neoplastic cells. Moreover, in contrast to β radiation, α particle-induced killing is independent of the oxygenation state or cell cycle during irradiation [22]. However, the main disadvantage of commonly used long-lived α emitters, such as ^225^Ac and ^223^Ra, is an escape of daughter radionuclides from radiopharmaceuticals. Since the energy of α decay is usually between 4 and 8 MeV, the daughter nuclide typically receives ~0.1 MeV of recoil energy. This is ~1000 times higher than the chemical binding energy; thus, the daughter radionuclide cannot be held by a chemical bond, and the efficient sequestration of daughter radionuclides in chelating ligands, such as cyclic or linear polyamino carboxylate chelators (DOTA or DTPA), is not possible [23]. Release of daughter radionuclides may lead to unwanted irradiation of healthy tissues that may cause severe radiotoxic effects [24]. Thus, so far, just a few α emitters have been investigated in preclinical studies for targeted mCRPC therapy, including bismuth-213, actinium-225, astatine-211, thorium-227, and lead-212. Up to date, the therapeutic efficiency and safety of alpha-targeted therapy of mCRCP have been tested in four clinical studies, including approximately 150 patients treated with PSMA small molecule inhibitor PSMA-617 labeled with actinium-225. The results have revealed that among all treated patients, almost 90% of them have shown PSA response in terms of any PSA decline with very low hematologic toxicity. One critical observation during these studies has been the high number of patients who discontinued therapy because of intolerable xerostomia. The first-in-human treatment concept with ^213^Bi-PSMA-617 has demonstrated a significant decrease in the PSA level without significant toxicities in mCRPC patients [25]. Despite promising results, the application of ^211^At, ^213^Bi, and ^225^Ac is very limited by the high costs of their production and low availability [26]. In contrast, the availability of radium ^223^Ra is significantly higher. The ^223^Ra can be produced in large amounts and relatively inexpensively from the ^227^Ac/^223^Ra generator, without significant impurities [27]. Because the decay of ^223^Ra takes place via several radioactive daughter nuclides, generating three additional α particles, a large amount of energy (~28 MeV) is deposited in the targeted organ. Therefore, ^223^Ra has the potential to deliver therapeutically relevant doses from a small amount of injected activity. Another attractive property of ^223^Ra radionuclide is a convenient, relatively long half-live (*t*_1/2_ = 11.43 days) that gives enough time for preparation, quality control, and shipment of the radiopharmaceuticals. The ^223^Ra in its simple form of radium dichloride ([^223^Ra]RaCl_2_) is the first α-particle emitting therapeutic agent approved by the U.S. Food and Drug Administration (FDA) for bone metastatic castration-resistant cancers. Surprisingly, despite these very attractive features, ^223^Ra has not yet found application in receptor-targeted therapy because Ra^2+^, similar to most alkali earth metals, does not form stable complexes with conventional bifunctional chelators. Additionally, as in the case of ^225^Ac, the release of radioactive decay products (^211^Bi and ^211^Pb) may cause their accumulation in several organs. To extend the use of ^223^Ra to applications other than bone metastases, several nanomaterials have been tested for stable incorporation of radium and linking to the targeting vector. Previous attempts to bind ^223^Ra within liposomes have not been successful because of their low stability in physiological fluids and relatively large diameter [28,29]. However, the next studies have shown better immobilization of ^223^Ra in magnetite nanoparticles [30], hydroxyapatites [31], barium sulfate nanoparticles [32], and CaCO_3_ microparticles [33]. In addition, the procedure of incorporation of ^223^Ra in NaA nanozeolites was developed by our group [34,35]. Unlike other proposed solutions, ^223^Ra labeling occurs by a Na^+^ for the Ra^2+^ ion exchange process after the synthesis of bioconjugate. In other systems, radioactive nanoparticles are first prepared and then modified, which causes the necessity of working with radioactive materials and loss of radioactivity by ^223^Ra decay. Promising results from these studies led us to design and evaluate a novel ^223^Ra-labeled, anti-PSMA targeted NaA nanozeolite for prostate cancer therapy. In this paper, we described the optimization of the synthesis of NaA nanozeolite to obtain a suitable carrier with a high ^223^Ra stability, as well as investigated targeting selectivity, the extent of internalization, and cytotoxicity of the radioimmunoconjugate in PSMA-expressing LNCaP C4-2 cells and PSMA-negative DU-145 cells.

## 2. Materials and Methods

### 2.1. Reagents

All chemicals were of analytical grade and used without further purification. Aluminum isopropoxide (≥98% purity) as Al source, LUDOX CL-X colloidal silica (45% suspension in water) as Si source, sodium hydroxide (≥97% purity) as cation source, and tetramethylammonium hydroxide (TMAOH) solution (25 wt. % in water) as template were purchased from Sigma Aldrich (St. Louis, MO, USA). Deionized water was obtained from Merck Millipore equipment (Burlington, MA, USA). All the above reagents were used for the synthesis of NaA zeolite powders. Epoxy silane functionalized polyethylene glycol (silane-PEG, 5000 Da) from Nanocs Inc. (New York, NY, USA) and ethanol (96% and 99.8% in water) from POCH (Gliwice, Poland) were used for the silanization process. 2-iminothiolane (2-IT), sodium bicarbonate (NaHCO_3_), ethylenediaminetetraacetic acid (EDTA), glycine, Ellman’s reagent, m-maleimidobenzoyl-N-hydroxysuccinimide ester (MBS), and dimethylformamide (DMF) from Sigma Aldrich (St. Louis, MO, USA) were used for the conjugation of anti-PSMA monoclonal antibody (D2B) to silane-PEG-nanozeolite. Radionuclide of ^223^Ra was isolated by a radiochemical separation from ^227^Ac source obtained from the Institute for Transuranium Elements (Karlsruhe, Germany) in the amount of 3 MBq. [^131^I] iodide with a specific activity of approximately 550 GBq mg^−1^ was supplied from the POLATOM Radioisotope Center (Świerk, Poland).

### 2.2. Synthesis of NaA Nanozeolites

The NaA nanozeolites were synthesized using the hydrothermal method according to the modified procedure of Piotrowska et al. [34] and Jafari et al. [36]. The aim of these modifications was to obtain NaA nanozeolites with a specific size (as small as possible), uniform shape, and specific surface characteristics (high surface area and narrow pore size distribution). To optimize the synthesis of NaA nanozeolites, the following conditions were tested: molar ratios of reagent mixture, time and temperature of the first stage of synthesis (crystals nucleation), time and temperature of the second stage of synthesis (crystals growth), and time and temperature of calcination process. The gel compositions and conditions of NaA nanozeolite synthesis, including aging time, crystallization temperature and time, crystal size, and crystallinity, are summarized in Table 1.

For the synthesis of NaA nanoparticles according to the procedures 1–4, the aluminosilicate gel was achieved by mixing freshly prepared homogeneous aluminate and silicate solutions. In the first step, 2.23 g sodium hydroxide was dissolved in 28.25 g deionized water, and 25.85 g of TMAOH was added. This solution was divided into two equal volumes. Then, 1.75 g of aluminum isopropoxide was added to one half of this solution and mixed on a magnetic stirrer until complete dissolution. A silicate solution was prepared by dissolving 4.31 g of colloidal silica in the other half of this solution and then was heated in an oven at 100 °C for 10 min to obtain a clear solution. In the next step, a silicate solution was slowly poured into the aluminate solution under high speed stirring to obtain a homogeneous gel. Stirring was performed at room temperature for 48 or 96 h. Upon stopping the aging process, the transparent mixture was placed in a high-pressure autoclave and heated at different temperatures (40 or 60 or 100 °C) for 5 or 24 h. Finally, the obtained material was washed several times with deionized water until the pH of the solution was neutral (pH ~ 7) and then dried at 80 °C overnight. In order to remove the template, the resulting product was calcined for 3 h at 500 °C.

For the synthesis of NaA nanoparticles according to procedures 5 and 6, sodium hydroxide (0.13 g) was dissolved in 19.68 g of distilled water, and afterward, 53.01 g of TMAOH was added. This sodium hydroxide solution was divided into two equal volumes. Then, 4.28 g of aluminum isopropoxide was added to one half of this solution and mixed on a magnetic stirrer until complete dissolution. A silicate solution was prepared by dissolving 8.0 g of colloidal silica in the other half of this solution and then was heated in an oven at 100 °C for 10 min to obtain a clear solution. In the next step, a silicate solution was slowly poured into the aluminate solution under high speed stirring to obtain a homogeneous gel. Stirring was performed in an ice bath for 96 h (aging time). After this time, the transparent mixture was placed in a high-pressure autoclave and heated at 100 °C for 24 h. Finally, the obtained material was washed several times with distilled water until the pH of the solution was neutral (pH ~ 7) and then dried at 80 °C overnight. In order to remove the template, the resulting product was calcined for 3 or 72 h at 500 or 600 °C.

### 2.3. Radiolabeling of NaA Nanozeolite with ^223^Ra Radionuclide

In the first step, the ^223^Ra radionuclide was obtained from the ^227^Ac/^227^Th/^223^Ra generator, where a long-lived ^227^Ac (*t*_1/2_ = 21.77 years) is deposited on extraction resin (Dipex-2 Actinide Resin, Eichrom Technologies LLC, Lisle, IL, USA) with daughter isotope ^227^Th (*t*_1/2_ = 18.7 days). After establishing the radioactive equilibrium, the ^223^Ra radionuclide was eluted from the resin using a 1 M HCl solution. In order to remove additional impurities, the ^223^RaCl_2_ solution was adsorbed on the second column filled with cationic resin (DOWEX 50WX4, 100–200 mesh), and ^223^Ra was eluted from the column with 8 mL of 5 M HNO_3_. The activity of collected fractions was measured using gamma-ray spectrometry. The eluate obtained from the generator was evaporated to dryness, and the remaining residue containing radioactivity was dissolved in 0.01 M HCl, evaporated to dryness, and dissolved in distilled water.

The ^223^Ra-labeled NaA nanozeolite was prepared by exchanging Na^+^ for ^223^Ra^2+^ cations. Briefly, 1 mg of NaA nanozeolite was suspended in 2 mL of RaCl_2_ solution (activity ~ 0.6 MBq) at pH 4 and sonicated in an ultrasound bath for 15 min. After shaking on a circular stirrer for 3 h at room temperature, the sample was centrifuged, and the supernatant was measured using gamma-ray spectrometry. The ^223^Ra-labeled nanozeolites were washed several times in distilled water to remove unbounded ^223^Ra isotope [34].

### 2.4. Surface Modification of NaA Nanozeolites with Silane–PEG

The modification procedure was carried out according to the method described by Hermanson [37]. Briefly, a solution containing 4% ethanol in water (*v*/*v*) was prepared, and the pH was adjusted to 5.0 with concentrated acetic acid. Then, 2 mg of silane–PEG-NH_2_ (5000 Da) was dissolved in acidic water/ethanol, and the solution was stirred for 5 min. After this time, 1 mg nanozeolite was added, and the mixture was stirred for 24 h at room temperature. The obtained product was washed several times with 99.8% ethanol to remove the excess of silane-PEG 5000 compound and dried at 110 °C for 30–40 min. The functionalization of the surface was carried out by using a silane coupling agent with three ethoxy groups and PEG molecules. The silane-PEG-NH_2_ agent was attached to the surface of nanozeolite NaA by the siloxane bonds formation. The surface modification of NaA nanozeolites with silane-PEG is presented in Figure 1.

### 2.5. Conjugation of Anti-PSMA Antibody to NaA-Silane-PEG Nanozeolite

The anti-PSMA monoclonal antibody D2B (IgG1) was prepared, as previously described by Frigeiro et al. [38]. Briefly, the D2B antibody was purified from a hybridoma culture supernatant by protein A affinity chromatography. The conjugation of the D2B antibody with the obtained NaA-silane-PEG nanozeolite consisted of three steps. In the first step, the antibody D2B-SH derivative was formed. For this purpose, 6 µL of 2-iminothiolane (2-IT) solution in water was added to the antibodies (concentration 1.5 mg/mL PBS) in the presence of 100 µL 1 M sodium bicarbonate (NaHCO_3_) and 10 µL EDTA 0.5 M in pH 8.0. Next, the reaction mixture was shaken for 2 h at room temperature and then overnight at 4 °C. After that, the reaction was blocked by adding 10 µL of 2 M glycine in water and incubated for 20 min at room temperature. The excess of 2-iminothiolane was separated by using PD-10 column GE-Healthcare. Afterward, the protein concentration was quantified with a spectrophotometer at 280 nm, and the free SH groups introduced with 2-IT by means of Ellman’s reagent. In the next step, 5 µL of m-maleimidobenzoyl-N-hydroxysuccinimide ester (MBS) solution in dimethylformamide (DMF) and 100 µL of 1 M NaHCO_3_ were added to the silane-PEG-NH_2_ functionalized nanozeolite in PBS buffer. The reaction mixture was incubated under shaking for 1.5 h at 30 °C. After this time, the reaction was blocked in the same way as in the first step by adding 10 µL of 2 M glycine. The excess of MBS reagent was separated from the solution by centrifugation (13,500 rpm for 10 min) and washed several times with PBS + EDTA buffer. Finally, the obtained NaA nanozeolite-silane-PEG-MBS was mixed with Abs-D2B-2-IT in a molar ratio of 1:16 in PBS + EDTA 4 mM environment. The conjugation was provided overnight at room temperature. After this time, to separate unbounded monoclonal antibody, the solution was centrifuged for 10 min at 13,500 rpm, then washed several times with PBS + EDTA buffer, and finally dispersed in 1 mL PBS + EDTA. The synthesis of ^223^RaA-silane-PEG-D2B radioimmunoconjugate is summarized in Figure 1.

### 2.6. High-Resolution Scanning Electron Microscopy (HR-SEM) with Energy Dispersive X-ray Spectroscopy (EDS)

The size and morphology of NaA zeolites were investigated by the high-resolution scanning electron microscopy Carl Zeiss “ULTRA plus” (Ultra-High-Resolution Imaging, Jena, Germany). The powdered NaA zeolite samples were fixed to a scanning electron microscopy holder with the Quick Drying Silver Paint (Agar Scientific Ltd., Essex, UK) conductive glue and coated with a thin layer of carbon using a vacuum evaporator (JEE-4X, JEOL, Tokyo, Japan) to assure conductivity and protect the sample from heat destruction. The elements presented in investigated samples were determined using the energy dispersive X-ray spectrometry (EDS) using Quantax 400 (Bruker, Billerica, MA, USA) microanalysis system. The mean size of NaA particles was determined from HR-SEM micrographs by measuring the diameters of about 100 particles/per point in 6 synthesized products.

### 2.7. Transmission Electron Microscopy (TEM)

For each sample, a small drop of the NaA nanozeolite-working solution was placed onto the transmission electron microscopy (TEM) copper mesh coated with a thin polymeric support film. After evaporation of the solvent under vacuum, the size and shape of the particles were analyzed by transmission electron microscope LEO 912AB (Zeiss, Jena, Germany) operating at an acceleration voltage of 120 kV.

### 2.8. XRD Diffraction Measurements

The identification and determination of the crystallinity of the nanozeolite were carried out by X-ray diffraction (XRD, D500 Diffractometer, Siemens) using a Vantec detector with Cu Kα radiation (1.548 Å) operated at voltage 40 kV and current 30 mA. The Rietveld method was applied using ICDD software for all powder patterns to confirm the structures of particles [39].

### 2.9. Fourier Transform Infrared Spectroscopy (FTIR) Analysis

Infrared spectra of the NaA zeolites were measured on Thermoscientific iS20 FT-IR Spectrometer, Thermo Scientifics. Spectra were collected in the mid (400–4000 cm^−1^) infrared region after 32 scans at 2 cm^−1^ resolutions. Samples were prepared using the standard KBr pellets method. To measure a spectrum, 1 mg of the NaA zeolite sample was well-mixed with 99 mg of KBr powder (1% sample in KBr) by grinding using an agate mortar and pestle. The obtained powder was then crushed in a mechanical die press to form a translucent wafer. Finally, the wafer was placed in an FTIR spectrometer for measurement. A pure KBr wafer was also made for the background corrections.

### 2.10. The Brunauer, Emmett, and Teller (BET) Measurements

To estimate the specific surface area of the synthesized nanoparticles, the samples were weighed and degassed. The degassing was carried out at 300 °C and about 200 mTorr for 24 h. After that, the samples were cooled to room temperature and inserted into sample ports of the ASAP 2420 MICROMETRICS instrument (Norcross, GA, USA). Nitrogen adsorption isotherms were measured at 77 K. The BET equation was used to calculate the specific surface areas [40].

### 2.11. Nanoparticle Tracking Analysis (NTA)

Nanoparticle tracking analysis (NTA) was performed with a NanoSight LM20 (NanoSight, Amesbury, UK), equipped with a sample chamber with a 640 nm laser. Working solutions of NaA zeolite particles were diluted 1:19 in the corresponding cell culture medium and injected in the sample chamber. All measurements were performed at room temperature. The hydrodynamic size distribution of the NaA zeolite samples was analyzed using the NTA 2.0 Build 127 software.

### 2.12. Zeta Potential and Polydispersity Index Measurements by Dynamic Light Scattering (DLS) Method

The zeta-potential and polydispersity index of the NaA zeolite samples were measured at 25 °C in a DTS 1067 capillary cell by using dynamic light scattering (DLS, Zetasizer NanoZS, Malvern, UK). Working solutions were diluted 1:8 in PBS and measured in triplicate with 20 sub runs. Zeta potentials were calculated using the Smoluchowski limit for the Henry equation with a setting calculated for practical use (*f(ka)* = 1.5). The polydispersity index (PDI) was obtained from the autocorrelation function. The default filter factor of 50% and the default lower threshold of 0.05 and an upper threshold of 0.01 were used.

### 2.13. Thermogravimetric Analysis

The thermal stabilities of unmodified NaA sample (NaA) and NaA sample modified with silane–PEG (NaA-silane-PEG) and with silane–PEG and D2B antibodies (NaA-silane-PEG-D2B) were determined by thermogravimetric analysis (TGA, Q500, TA Instruments, New Castle, DE, USA). During thermogravimetric measurements, the tested samples were heated to a temperature of 800 °C, maintaining a heating rate of 10 °C/min, in an inert gas-nitrogen atmosphere.

### 2.14. Analysis of the Stability of Radioimmunoconjugate

A portion of radioimmunoconjugate (100 µL which represents 100 µg/mL and 0.065 MBq) was placed in a dialysis tube (D-Tube Dialyzer Midi with 3.5 kDa cut-off membrane, Novagen) and was dialyzed against 20 mL of 0.9% NaCl or 0.01 M PBS for up to 12 days at room temperature. Every 24 h, a 1 mL aliquot was taken and measured on a γ-spectrometer containing Coaxial High Purity Germanium (HPGe) detector (GX 1080) connected to multichannel analyzer DSA-1000 and Genie 2000 software (Canberra, Meriden, CT, USA). The percentage of liberated activity of ^223^Ra, ^211^Bi, and ^211^Pb was determined by their characteristic gamma-lines (269.4 keV 13.6% for ^223^Ra, 404.9 3.8% and/or 831.9 keV 3.8% for ^211^Pb, and 350.1 keV 12.8% for ^211^Bi). To determine stability in human serum (HS), 100 µL of radioimmunoconjugate (which represents 100 µg/mL and 0.065 MBq) was added to 1 mL of human serum, vortexed, and incubated at 37 °C up to 12 days. Every 24 h, a 1 mL aliquot was taken and measured on a γ-spectrometer.

### 2.15. Cell Cultures

The human epithelial prostate carcinoma cell line DU-145 (ATCC^®^HTB-81™) and the human lymph node prostate carcinoma cell line LNCaP C4-2 (ATCC^®^ CRL-3314^™^) were purchased from the American Type Tissue Culture Collection (ATCC, Rockville, MD, USA) and maintained according to the ATCC protocols. Briefly, DU-145 was cultured in EMEM medium supplemented with 10% FBS, 2 mM L-glutamine, and 100 U/mL penicillin-streptomycin, whereas LNCaP C4-2 was grown in RPMI 1640 medium containing 10% FBS, 2 mM L-glutamine, 10 mM HEPES, 40 mg/l folic acid, and 100 U/mL penicillin-streptomycin. Both cell lines were maintained in an incubator at 37 °C with 5% CO_2_.

### 2.16. Determination of the Number of Attached Anti-PSMA Antibodies Per NaA Nanozeolite

The average number of D2B molecules attached to one NaA nanozeolite particle was determined by the radiometric method. This method relied on the conjugation of radioiodinated protein to nanoparticles by using the procedure described by Cędrowska et al. [41], with modifications necessary for the D2B antibody. Briefly, 0.5 mg of thiol-derivatized anti-PSMA mAb (D2B-SH) in 0.05 M PBS/10 mM EDTA was labeled with ^131^I (20–40 MBq) in the presence of Iodogen. The radioiodinated ^131^I-D2B-SH was purified on a PD-10 column filled with Sephadex G-25 resin (GE Healthcare Life Sciences, Piscataway, NJ, USA). Next, 50 μg of ^131^I-D2B-SH was added to 1 mg of maleimide-modified nanozeolite (MBS-NaA), and the coupling reaction was performed as described in detail above. In the next step, ^131^I-D2B-silane-PEG-NaA was separated from the solution by centrifugation (13,500 rpm for 10 min), washed several times with PBS/EDTA, and finally dispersed in 0.1 mL PBS. The radioactivity of each fraction was measured by γ-spectrometry, and the conjugation yield of ^131^I-D2B-SH to MBS-NaA was determined from the proportion of radioactivity coupled to nanoparticles. The number of ^131^I-D2B-SH attached to each MBS-NaA was calculated by dividing the moles of ^131^I-D2B-SH bound to nanoparticles by the moles of used MBS-NaA. Thus, the formed ^131^I-D2B-PEG-silane-NaA was used in the initial in vitro studies to determine its specificity towards PSMA-receptor and internalization properties.

### 2.17. Binding Specificity and Internalization Properties of ^131^I-D2B-PEG-Silane-NaA

The specificity of binding of ^131^I-D2B-PEG-silane-NaA to PSMA-receptor was evaluated on LNCaP C4-2 and DU-145 cells using a similar protocol, as described by Pruszyński et al. [42]. Briefly, both cell lines (2 × 10^5^ cells per well) were incubated with ^131^I-D2B-PEG-silane-NaA, either alone or with an addition of 25-fold molar excess of unlabeled D2B antibodies to block the receptors for 2 h at 4 °C. Then, the cells were washed twice with cold PBS and lysed by the addition of twice 0.5 mL 1 M NaOH for 10 min at 37 °C. The lysates were counted for radioactivity in an automatic γ-counter (Wizard 2480, Perkin-Elmer). The intracellular retention of ^131^I-PSMA-PEG-silane-NaA was evaluated by incubation with LNCaP C4-2 (2 × 10^5^ cells per well) at 4 °C for 1 h. A 25-fold molar excess of unlabeled D2B antibodies was added in parallel to assess non-specific binding. Next, the unbound fraction was washed away, and the cells were supplemented with fresh medium and incubated at 37 °C for 1 and 24 h. After incubation, the medium fraction was collected (supernatant) prior to an acid wash using 0.05 M glycine-HCl (pH 2.8), through which the membrane-bound fraction was collected. Finally, the cells were lysed by the addition of twice of NaOH (internalized fraction). The radioactivity of fractions collected during cell studies was detected using Wizard 2^®^ automatic gamma counter 2480 (Perkin Elmer, MA, USA). The experiments were performed in triplicates, and the results are presented as mean ± standard deviation (SD).

### 2.18. Measurements of Metabolic Activity by the MTT Assay

The effects of immunoconjugate NaA-silane-PEG-D2B, radioimmunoconjugate ^223^RaA-silane-PEG-D2B, and ^223^RaCl_2_ on the metabolic activity of DU-145 and LNCaP C4-2 cells were measured with 3-(4,5-dimethyl-2-thiazolyl)-2,5-diphenyl-2H-tetrazolium bromide (MTT, Sigma Aldrich) assay. Cells were seeded in 96-well plates (TTP) at a density of 2.5 × 10^3^ cells/well and 10 × 10^3^ cells/well for DU-145 and LNCaP C4-2 cell line, respectively. After 24 h, the cells were treated with increasing concentrations of NaA-silane-PEG-D2B (1.56–100 µg/mL), ^223^RaA-silane-PEG-D2B (0.31–20 kBq/mL and 0.94–60 kBq/mL for LNCaP C4-2 and DU-145, respectively), and ^223^RaCl_2_ (0.31–20 kBq/mL and 0.94–60 kBq/mL for LNCaP C4-2 and DU-145, respectively) for 48, 72, and 96 h. After incubation, the medium was removed, and the cells were washed with PBS and incubated with 100 µL/well of MTT solution (0.1 mg/mL) in the medium for 3 h at 37 °C. After this time, the dye solution was removed, and the formazan crystals were dissolved in DMSO. The absorbance was measured at 570 nm in plate reader spectrophotometer Infinite M200 (Tecan, Grödig, Austria). At least three independent experiments in six replicate wells were conducted per point. The metabolic activity of cells treated with a tested compound (in percent of metabolic active cells compared with control cells that were untreated) was calculated from the absorbance measurements. The values of half-maximal inhibition concentrations (*IC*_50_) were estimated from the survival curves.

### 2.19. Statistical Analysis

Statistical analysis of the obtained data was performed using Statistica 7.1 software (Stat Soft. Inc., Tulsa, OK, USA). The data are expressed as mean ± standard deviation (SD) of at least three independent experiments. For the cytotoxicity studies, the data were compared using an unpaired t-test with a significance level of *p* < 0.05.

## 3. Results and Discussion

### 3.1. Influence of Gel Composition and Synthesis Conditions on the Size, Shape, and Crystallinity of Synthesized Zeolites

Preparation of a high-quality carrier of a radionuclide with small particle size, narrow particle size distribution, and high surface area with large pore volume is of great importance for the synthesis of nanoparticle-based radiobioconjugates. Among different nanoparticles, which can contain radionuclides, the NaA nanozeolite possesses favorable physical properties, such as high microporosity, the potential to be functionalized, and very low or no toxicity [34,43,44]. Up to now, just a few studies have been performed to control the size of NaA zeolites crystals in the presence of an organic template [34,36,45,46,47,48,49,50,51]. These studies have recognized that several synthesis parameters, such as gel composition, aging time, temperature, alkalinity (pH and content of cation), hydrothermal reaction temperature, nature of reactants, and composition of reaction mixtures, affect the crystallization routes and kinetics in zeolites synthesis. Hence, the first task of the current work was to select an optimal procedure for the synthesis of NaA nanozeolite as the ^223^Ra radionuclide carrier for targeting of PCa. We used six different procedures for the synthesis of zeolites. The products of these syntheses were characterized at this stage of work by scanning electron microscopy (SEM) and XRD diffraction (Table 2 and Figure 2). As presented in Table 2, the average diameters of particles from sample 1 and sample 2 were similar (i.e., the diameter of about 630 nm and 550 nm, respectively). These particles had irregular shape with visible impurities (Figure 2A,B). The XRD patterns of sample 1 and sample 2 showed 80% and 65% of crystallinity, respectively. These results revealed that aging time and crystallization time had no influence on size but affected percent of the crystallinity of samples. The results showed also that crystallization temperature played an essential role in the synthesis of zeolite particles (samples 3 and 4). The decreasing temperature of the process from 100 °C to 60 °C or 40 °C resulted in the formation of a product with low crystallinity or amorphous structures. This result was in line with data published by Cundy et al. [52] and Mirfendereski et al. [51], who reported that crystallization temperature had a strong effect on the formation of zeolites and that increasing synthesis temperature significantly increased the crystallinity of zeolite NaA.

As presented in Figure 2C,D, zeolite particles from sample 3 and sample 4 were highly non-uniform in shape. The particles from sample 3 and sample 4 had a mean particle size of ~90 nm and~ 130 nm, respectively. The XRD spectra of these samples confirmed the HR-SEM results. The XRD pattern of sample 3 revealed a broad diffuse halo at around 2*θ* = 23°, with no detectable sharp diffraction peaks corresponding to any crystalline phase. This result indicated that the sample 3 was amorphous. The broadening of the peak for amorphous silica at *2θ* = 23.5° was also reported by others [53,54,55]. The XRD pattern of sample 4 showed 30% of crystallinity.

According to the literature [51,56], the SiO_2_/Al_2_O_3_ ratio in the reaction system plays an important role in determining the structure and composition of synthesized crystals. Hence, we changed the composition of the initial reactive mixture to improve the crystallinity of zeolite particles and to decrease their size. As presented in Figure 2E, the morphology of zeolite crystals from sample 5 was cubic-like, and no remaining amorphous materials were detected by this technique, which indicated a high crystallization degree. The XRD pattern of sample 5 confirmed the HR-SEM results and showed 94% of the crystallinity of this sample. Unfortunately, the average diameter of these particles was too big for our purpose (~240 nm). As known, an aging step prior to crystallization strongly affects crystal size by an increase in a number of nuclei [45,57,58]. It has been reported that a decrease in aging temperature and time results in substantially smaller NaA zeolite particles [59,60,61]. Hence, to obtain smaller crystals with homogeneous morphology and high crystallization degree, we decreased the temperature of the aging process from room temperature to the ice bath temperature. As illustrated in Figure 2F, the average diameter of obtained cubic-like crystals decreased to ~120 nm. The XRD pattern of sample 6 showed 95% of crystallinity with no indication of phase impurities or amorphous material. Since it has been reported that only nanocarriers of size ≤ 150 nm are able to enter or exit fenestrated capillaries in the tumor microenvironment or liver endothelium, as well as do not easily leave the capillaries that perfuse tissues, such as the kidney, heart, and lung [62,63], the NaA nanozeolite obtained according to the last procedure (sample 6) was selected for further physicochemical analysis.

### 3.2. Physicochemical Characteristics of the Chosen NaA Nanozeolite (Sample 6)

The structure of synthesized NaA zeolites (sample 6) was elucidated by transmission electron microscopy (TEM). The representative TEM image at high magnification confirmed the particle size distribution similar to this obtained by HR-SEM and indicated the presence of nanocrystals with regular cubic-like shape and an average diameter of 120 nm (Figure 3A).

According to the XRD diffractogram, the obtained material presented only one phase of a pure zeolite type NaA (Figure 3B). The peaks characterizing the NaA zeolite at *2θ* = 10.24, 12.56, 16.24, 21.86, 24.20, 26.36, 27.36, 30.22, 34.50 were the same as peaks reported by Treacy and Higgins [64], suggesting successful synthesis of zeolite NaA with good crystallinity. The presence of sharp diffraction peaks indicated high crystallinity of the obtained nanoparticles and corresponded with the NaA framework. Based on the Rietveld refinement, the crystallinity was calculated as 95%.

The FTIR spectrum of sample 6 is presented in Figure 3C. We observed the characteristic broadband at about 3423 cm^−1^, and band around 1646 cm^−1^ originated from stretching vibrations and bending vibrations of O–H bonds, respectively, which took place in water adsorbed from the air after the process of calcination. The band about 1000 cm^−1^ was associated with asymmetrical vibration of Si–O–Si bonds from the tetrahedral layer. The bands below 1000 cm^−1^ were related to the T–O–T bending of vibration mode (T = Al, Si). This result was in agreement with the observations of others [39,61,65], confirming the formation of double four-rings of the zeolite NaA framework.

The EDS studies confirmed that the sample 6 was a pure NaA product, composed of oxygen (*A*% = 59.08%), silica (*A*% = 17.69%), aluminum (*A*% = 13.05%), and sodium (*A*% = 10.18%) (Figure 2D). The ratio between oxygen, aluminum, and silica O:(Al + Si) was 2:1, which is characteristic of crystalline aluminosilicates [66]. The carbon peak was attributed to the carbon adhesive used to hold the mesh on to the SEM specimen holder.

The surface area, average pore diameter, cumulative pore volume, micropore volume (pores < 2 nm), and microporosity were measured by BET nitrogen adsorption-desorption isotherms and BJH (Barrett, Joyner, and Halenda) pore size distribution analysis (Figure 4, Table 3). As shown in Figure 4A, the nitrogen adsorption isotherms of NaA nanozeolite calcinated at 500 °C for 3 h and NaA nanozeolite calcinated at 600 °C for 72 h presented characteristic features of the type IV class with respect to IUPAC adsorption-desorption classification with a distinct H4 of the hysteresis loop, indicating the combination of micropores and mesopores in the solids [67]. Both isotherms exhibited a nitrogen uptake at low relative pressures (<0.01), indicative of the characteristics of micropores. The results of pore size distribution analysis of these samples confirmed the presence of micropores (peaks at 2.58 nm and 2.88 nm) and mesopores (peaks at 44.4 nm and 43.6 nm) (Figure 4B). As presented in Table 3, the NaA nanozeolite calcinated at 600 °C for 72 h had a higher surface area (485.03 m^2/^g vs. 428.52 m^2/^g), total pore volume (0.35 cm^3^/g vs. 0.27 cm^3^/g), and micropore volume (0.217 cm^3^/g vs. 0.179 cm^3^/g) than the NaA nanozeolite calcinated at 500 °C for 3 h. Hence, these results revealed that the increased time and temperature of calcination improved the removal of the organic template from NaA nanozeolite pores. This affected the increased surface sorption properties (data not shown).

### 3.3. Physicochemical Characteristics of the NaA Nanozeolite Modified with Silane-PEG and Anti-PSMA Antibody D2B

The TGA analysis was used to confirm the modification of NaA nanozeolite with silane-PEG and anti-PSMA D2B antibody. The analysis of TGA thermograms of unmodified NaA nanozeolite (NaA), silane-PEG molecules (silane-PEG), NaA nanozeolite functionalized with silane-PEG (NaA-silane-PEG), and NaA nanozeolite functionalized with silane-PEG and anti-PSMA D2B antibody (NaA-silane-PEG-D2B) is presented in Figure 5. In the case of NaA, the loss of mass (~20%) occurred in the range from 30 °C to 150 °C, which was credited to evaporation of water molecules, adsorbed on their surface and inside the pores. Water evaporation was also observed for other investigated samples, but at higher temperatures because of the silane-PEG and D2B molecules modifying NaA nanozeolite. Degradation and evaporation of the silane-PEG occurred in the range from 350 °C to 450 °C. The thermogravimetric curve of the NaA-silane-PEG revealed a marked loss of mass (~13%) in the range from 200 °C to 450 °C. This effect could be attributed to the degradation of the silane-PEG molecules on NaA zeolite because it took place at the same conditions as earlier was observed in the case of silane-PEG molecules. The consecutive loss of mass (about 1%) was observed in the case of NaA-silane-PEG-D2B, which could be attributed to the evaporation of D2B monoclonal antibody degradation products.

The NanoSight studies of NaA nanozeolite (NaA), NaA nanozeolite functionalized with silane-PEG (NaA-silane-PEG) and functionalized with silane-PEG and D2B antibody (NaA-silane-PEG-D2B) confirmed the TGA analysis, providing experimental evidence for the presence of additional molecules on the NaA nanozeolite surface. As presented in Table 4, the pure NaA had a hydrodynamic diameter of 196.7 ± 53.5 nm. The addition of silane-PEG molecules increased this diameter to 216.9 ± 56.6. The hydrodynamic diameter was further increased with the addition of the D2B antibody to 225.1 ± 48.6 nm. These results were in good agreement with the findings presented by other authors, who reported that proteins or other molecules adsorbed onto nanoparticle surfaces increased their size [68,69].

As known from the other publications [70,71], hydrodynamic size of NaA measured by NanoSight appeared larger than the size observed on HR-SEM and TEM images (~200 nm vs. 120 nm). We assumed that the reason behind this was the solvent layer and any coating material from the RPMI 1640 or EMEM cell culture medium attached to the particle, known as a protein corona. It is well established that when the dispersed particle moves through a liquid medium, a thin electric dipole layer of a solvent adheres to its surface. This layer facilitates the adsorption of proteins on nanoparticles, which can modify the diverse physicochemical properties of nanoparticles, such as size, surface composition, and surface charge [72]. The surface charge of these compounds, determined by the DLS method, revealed the high negative value of zeta potential of the NaA (−40.3 ± 2.11 mV), NaA-silane-PEG (−43.1 ± 2.45 mV), and NaA-silane-PEG-D2B (−43.1 ± 2.45 mV), showing their high stability. According to the literature, the zeta potential value from −30 mV to +30 mV is considered optimum for the good stabilization of a nanodispersion. However, a large positive or negative value of zeta potential indicates good physical stability of nanosuspensions due to electrostatic repulsion of individual particles, which are less prone to form aggregates or increase in particle size [73,74]. In addition, the analysis of polydispersity indices (PDI) of all compounds revealed that the PDI values ranged from 0.292 to 0.186, which denoted a relatively monodisperse distribution system. The numerical value of PDI ranges from 0 (for a perfectly uniform sample with respect to the particle size) to 1.0 (for a highly polydisperse sample with multiple particle size populations). In drug delivery applications, a PDI of 0.3 and below is considered to be acceptable [75].

### 3.4. Stability of Radioimmunoconjugate ^223^RaA-Silane-PEG-D2B

The NaA nanozeolite was labeled with 0.65 MBq of ^223^Ra^2+^ radionuclide with a high yield of 99.8 ± 1.8%. The stability of ^223^RaA-silane-PEG-D2B in physiological salt (0.9% NaCl) and human blood serum after 12 days are presented in Figure 6.

A leakage of mother radionuclide ^223^Ra and its decay products ^211^Bi and ^211^Pb from ^223^RaA-silane-PEG-D2B was measured on high resolution (HPGe) γ-spectrometer. The radiometric analysis showed that leakage of ^223^Ra from bioconjugate was 3% in 0.9% NaCl and 4% in human blood serum after 12 days. At the same time, the release of ^211^Pb and ^211^Bi in 0.9% NaCl and human serum was below 6%.

This retention was higher than expected for the size of the nanozeolites and was explained by the resorption of decay product ^219^Rn on the nanoparticle. It is well known that zeolite A, as a molecular sieve, exhibits adsorption ability for gaseous radon isotopes [76]. However, according to Holzwarth et al. [77], the results obtained using nanozeolite labeled with ^223^Ra equilibrated with human serum could not be transferred to in vivo models. Blood flow may rapidly dislocate the decay products from the surface of the nanozeolite particles and reduce the resorption probability, but considering the internalization rate of the bioconjugate inside the cell, perhaps after reaching target cells, the resorption process will play an important role in preventing the release of free ^211^Pb and ^211^Bi from the cells. It is noteworthy that, unlike ^225^Ac, the release of the decay products from ^223^Ra is a negligible problem because 75% of the α particle’s energy is emitted within 4 s of the ^223^Ra decay.

### 3.5. The Average Number of Attached Anti-PSMA D2B mAbs Per NaA Nanoparticle

Based on a radiometric method, in which thiol-derivatized ^131^I-D2B-SH was reacted with maleimide-functionalized nanozeolite (MBS-NaA), the coupling efficiency was estimated by measuring the proportion of radioactivity bound to nanoparticles in relation to initially added. The number of ^131^I-D2B-SH attached to one MBS-NaA was calculated by dividing the moles of ^131^I-D2B-SH bound to nanoparticles by the moles of used MBS-NaA. It was estimated that ~ 50 D2B molecules were coupled with one NaA particle.

### 3.6. Binding Specificity and Internalization Properties of ^131^I-D2B-PEG-Silane-NaA

The performed studies revealed that ^131^I-D2B-PEG-silane-NaA specifically bound to PSMA-receptor (Figure 7A). The percentage of the radioactive fraction that was bound to PSMA-positive LNCaP C4-2 cells was as high as 19.7% ± 2.4%. Blocking of cells’ antigens with a 25-fold molar excess of unlabeled D2B mAb reduced binding by the factor of three (radioactive fraction bound was 6.6% ± 1.3%, *p* < 0.005). In line with this, PSMA-negative DU-145 cells showed only very low unspecific binding (1.1% ± 0.2%, *p* < 0.005) that confirmed the specific binding of ^131^I-PSMA-PEG-silane-NaA to PSMA-receptor.

The ^131^I-D2B-PEG-silane-NaA radiobioconjugate was readily internalized into LNCaP C4-2, as after 1 h of incubation at 37 °C, almost 84.2% ± 4.1% of initially bound radioactivity was inside the cells (Figure 7B), although it slightly decreased to 54.3% ± 2.2% at 24 h. The observed behavior indicated that internalized radiobioconjugate was likely subjected to lysosomal degradation inside the cell, and the formed radioactive catabolites were excreted into the supernatant (39.8% ± 1.2% at 24 h), as noticed and confirmed in previous studies [78,79].

### 3.7. Effect of the Synthesized Compounds on Cells’ Metabolic Activity

The cytotoxic effect of bioconjugate NaA-silane-PEG-D2B, radioimmunoconjugate ^223^RaA-silane-PEG-D2B, and ^223^RaCl_2_ was investigated in the PSMA-positive LNCaP C4-2 cells and in the PSMA-negative DU-145 cells. As presented in Figure 8, the metabolic activity of LNCaP C4-2 cells and DU-145 cells exposed to bioconjugate NaA-silane-PEG-D2B, radioimmunoconjugate ^223^RaA-silane-PEG-D2B, and ^223^RaCl_2_ varied significantly. Immunoconjugate NaA-silane-PEG-D2B at concentrations from 1.6 to 100 µg/mL was nontoxic in DU-145 (Figure 8A) and LNCaP C4-2 cells (Figure 8B) after 48, 72, and 96 h. There was no significant difference in metabolic activity between these two cell lines (*p* = 0.732).

In contrast, the treatment of cells with radioimmunoconjugate ^223^RaA-silane-PEG-D2B or ^223^RaCl_2_ resulted in a significant reduction of metabolic activity in a concentration- and time-dependent manner. As shown in Figure 8C, the treatment of DU-145 cells with ^223^RaA-silane-PEG-D2B significantly decreased their metabolic activity after 48 h starting from the concentration of 1.87 kBq/mL (~7% reduction) and reaching a value of 41% reduction at the concentration of 60 kBq/mL, after 72 h starting from the concentration of 0.94 kBq/mL (~15% reduction) and reaching a value of 66% reduction at the concentration of 60 kBq/mL, and after 96 h starting from the concentration of 0.94 kBq/mL (~25% reduction) and reaching a value of 72% reduction at the concentration of 60 kBq/mL.

The recorded values of the metabolic activity of DU-145 cells exposed to ^223^RaCl_2_ also gradually decreased in a concentration- and time-dependent manner. However, the reduction of metabolic activity was smaller than that for the radioimmunoconjugate ^223^RaA-silane-PEG-D2B. As presented in Figure 8E, the treatment of DU-145 cells with ^223^RaCl_2_ significantly decreased their metabolic activity after 48 h starting from the concentration of 30 kBq/mL (7.74% reduction) and reaching a value of 11.87% reduction at the concentration of 60 kBq/mL, after 72 h starting from the concentration of 7.5 kBq/mL (5.66% reduction) and reaching a value of 28.29% reduction at the concentration of 60 kBq/mL, and after 96 h starting from the concentration of 1.87 kBq/mL (6.80% reduction) and reaching a value of 50.79% reduction at the concentration of 60 kBq/mL. There were significant differences in metabolic activity between DU-145 cells treated with ^223^RaCl_2_ and with radioimmunoconjugate ^223^RaA-silane-PEG-D2B after 48 h (*p* = 0.0012), 72 h (*p* = 0.0010), and 96 h (*p* = 0.0003). The assessed *IC*_50_ values for ^223^RaA-silane-PEG-D2B and for ^223^RaCl_2_ after 96 h were 7.6 kBq/mL and 37.4 kBq/mL, respectively.

A similar trend in a more pronounced manner was observed in LNCaP C4-2 cells. As presented in Figure 8D the treatment of LNCaP C4-2 cells with ^223^RaA-silane-PEG-D2B significantly decreased their metabolic activity after 48 h, starting from the concentration of 0.31 kBq/mL (4.86% reduction) and reaching a value of 46.04% reduction at the concentration of 20 kBq/mL, after 72 h starting from the concentration of 0.31 kBq/mL (22% reduction) and reaching a value of 70.71% reduction at the concentration of 20 kBq/mL, and after 96 h starting from the concentration of 0.31 kBq/mL (30.59% reduction) and reaching a value of 75.79% reduction at the concentration of 20 kBq/mL. As presented in Figure 8F the treatment of LNCaP C4-2 cells with ^223^RaCl_2_ significantly decreased their metabolic activity after 48 h, starting from the concentration of 10 kBq/mL (3.87% reduction) and reaching a value of 5.82% reduction at the concentration of 20 kBq/mL, after 72 h starting from the concentration of 10 kBq/mL (15.48% reduction) and reaching a value of 24.69% reduction at the concentration of 20 kBq/mL, and after 96 h starting from the concentration of 0.625 kBq/mL (7.80% reduction) and reaching a value of 53.75% reduction at the concentration of 20 kBq/mL.

The comparative analysis of MTT results between LNCaP C4-2 and DU-145 cell lines revealed that LNCaP C4-2 cells were nearly four times more sensitive to both radiocompounds ^223^RaA-silane-PEG-D2B and ^223^RaCl_2_ than DU-145 cells (*IC*_50_ 2 kBq/mL vs. 7.6 kBq/mL and *IC*_50_ 9.3 kBq/mL vs. 37.4 kBq/mL after 96 h, respectively). This result was in corroboration with the earlier observation that the LNCaP C4-2 cells are more radiosensitive than DU-145 cells [80]. This effect could, in part, be attributed to the p53 status of these cell lines because this protein is one of the key molecules involved in a cell response to ionizing radiation [81]. The LNCaP C4-2 cells are p53-positive [82], while in contrast, the DU-145 cells are p53-mutant cells [83]. There are a number of papers covering the role that the p53 plays in determining the fate of solid tumor-derived cells after exposure to ionizing radiation, leading to the conclusion that loss of p53 function is responsible for radioresistance [84,85,86]. However, there are many differences between LNCaP and DU-145 cells, aside from p53 status, which may be responsible for different radiosensitivity [87]. A conflicting report has indicated that the responses of DU-145 cells to beta particles, alpha particles, or gamma rays are almost identical, as for LNCaP cells [88].

Our results revealed also that the LNCaP C4-2 cells were nearly four times more sensitive to radioimmunoconjugate ^223^RaA-silane-PEG-D2B than DU-145 cells. This finding suggested the enhanced internalization of targeted radioimmunoconjugate by the PSMA receptors on the surface of LNCaP C4-2 cells. It is already known that PSMA undergoes constitutive internalization, and this process is significantly accelerated when this molecule is bound by an antibody [13]. The LNCaP C4-2 cells overexpress the PSMA protein cells [89], while, in contrast, the PSMA receptor is not expressed on the surface of DU-145 cells [90]. This observation was supported by our results dealing with binding specificity and internalization properties of ^131^I-D2B-PEG-silane-NaA, which indicated that the ^131^I-D2B-PEG-NaA radiobioconjugate was highly internalized into LNCaP C4-2 but not into DU-145 cells. Unexpectedly, we also observed a toxic effect of radioimmunoconjugate in DU-145 cells. This increase in radiotoxicity was significantly less than that of LNCaP C4-2 cells under the same culture conditions. We assumed that this effect might be probably related to a long-term incubation process, in which the radioimmunoconjugate is sedimented directly on the cell surface. This non-specific process that could increase the amount of radioactivity delivered into DU-145 cells was already observed for other nanoparticle-based radioimmunoconjugates [41].

## 4. Conclusions

In conclusion, a novel nanoparticle-based radioimmunoconjugate ^223^RaA-silane-PEG-D2B was successfully synthesized and characterized. The procedures for NaA nanozeolite synthesis, its modification with silane-PEG molecules, functionalization with anti-PSMA D2B antibody, and radiolabeling reaction were efficient and strictly dependent on the reaction conditions employed. Extensive physicochemical characterization of this compound revealed that the NaA nanozeolite, modified with silane-PEG and functionalized with anti-PSMA D2B antibody could be used as a carrier of ^223^Ra radioisotope and daughter radionuclides with a very high yield of stability (>95% up to 12 days). The competition binding studies confirmed a high-affinity binding of this radioimmunoconjugate towards the PSMA receptor and its very fast and selective internalization into PSMA-positive LNCaP C4-2 cells, but not into PSMA-negative DU-145 cells. In addition, the analysis of cytotoxicity of this radioimmunoconjugate by the MTT assay confirmed that this conjugate was about four-fold more toxic in LNCaP C4-2 cells than in DU-145 cells. The results of the present study suggested that the ^223^RaA-silane-PEG-D2B might be a promising agent for the targeted treatment of human PCa. Further preclinical in vitro and in vivo studies in athymic nude mice bearing subcutaneous PSMA-expressing prostate cancer xenografts are underway in our laboratory.

## 5. Patents

A. Kasperek, A. Bilewicz, Therapeutic radiopharmaceutical labeled with radionuclides of radium and method for its obtaining, Polish Patent no 217466, 2014.

## Figures and Tables

**Figure 1 materials-13-03875-f001:**
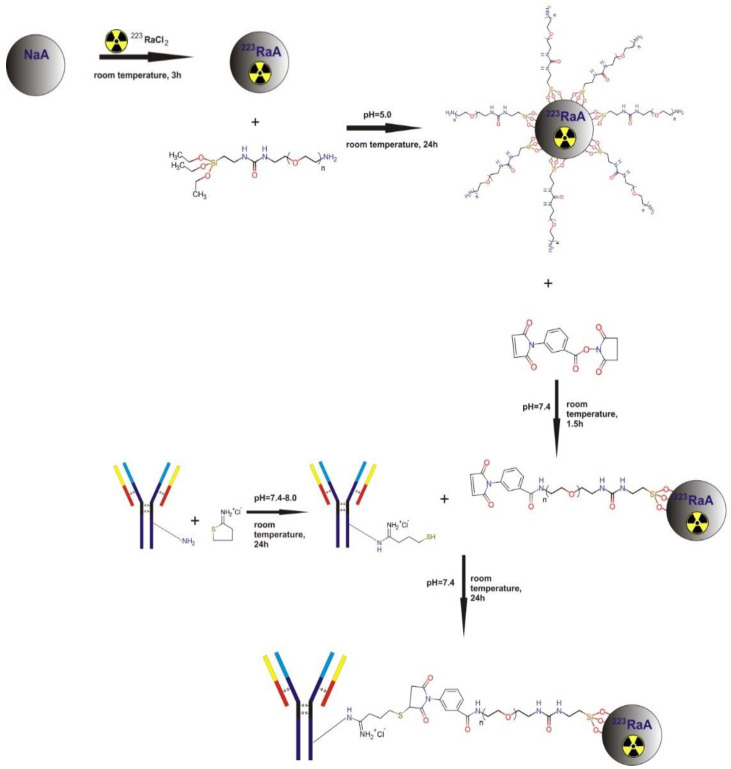
Steps of the ^223^RaA-silane-PEG-D2B radiobioconjugate synthesis.

**Figure 2 materials-13-03875-f002:**
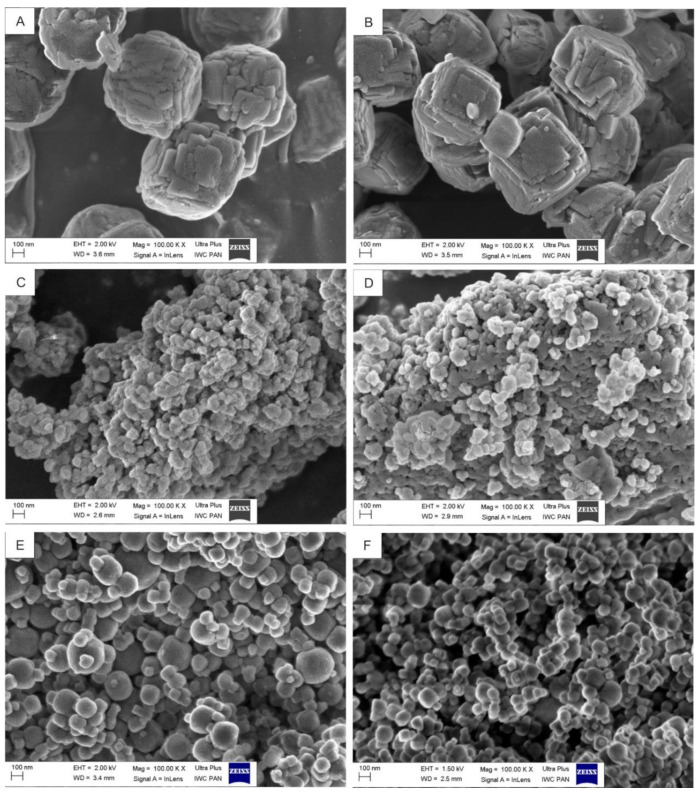
Representative scanning electron microscopy images of 6 NaA particle samples synthesized using different gel compositions and conditions of synthesis. (**A**) sample 1, (**B**) sample 2, (**C**) sample 3, (**D**) sample 4, (**E**) sample 5, (**F**) sample 6.

**Figure 3 materials-13-03875-f003:**
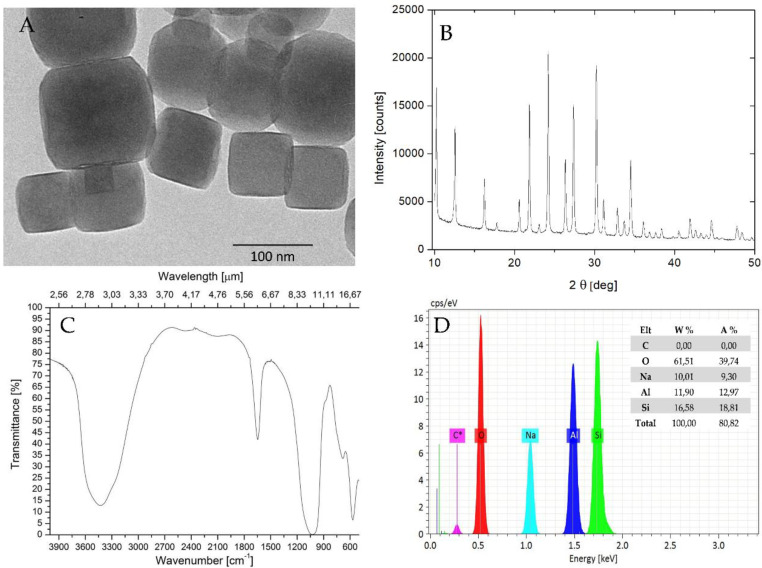
Physicochemical characteristics of NaA nanozeolites (sample 6). (**A**) Transmission electron microscopy analysis, (**B**) XRD analysis, (**C**) FT-IR spectra, (**D**) EDS analysis.

**Figure 4 materials-13-03875-f004:**
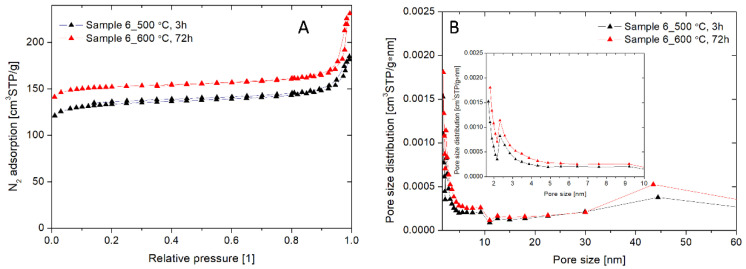
The effect of time and temperature of calcination on porous parameters of NaA nanozeolite (sample 6). (**A**) low-temperature nitrogen adsorption isotherms, (**B**) pore size distributions. Inset shows pore size distributions in the range from 0 to 10 nm.

**Figure 5 materials-13-03875-f005:**
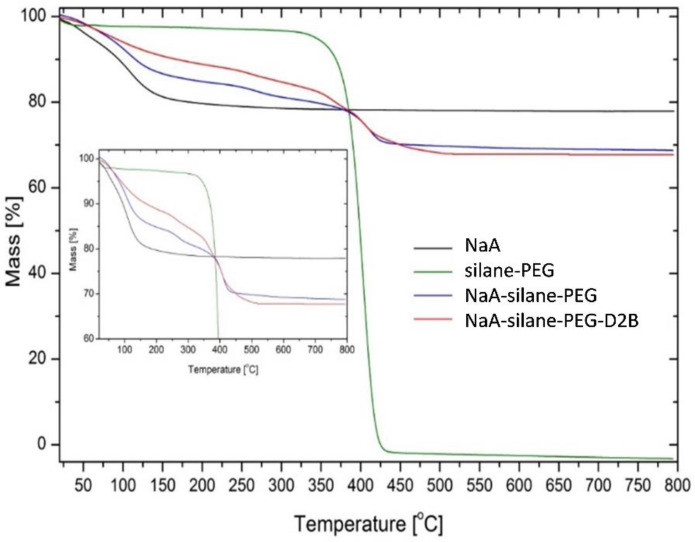
Representative thermograms of NaA nanozeolite (NaA), silane-PEG molecules (silane-PEG), NaA nanozeolite functionalized with silane-PEG (NaA-silane-PEG), and NaA nanozeolite functionalized with silane-PEG and D2B antibody (NaA-silane-PEG-D2B). Inset shows the values of mass from 60–100%.

**Figure 6 materials-13-03875-f006:**
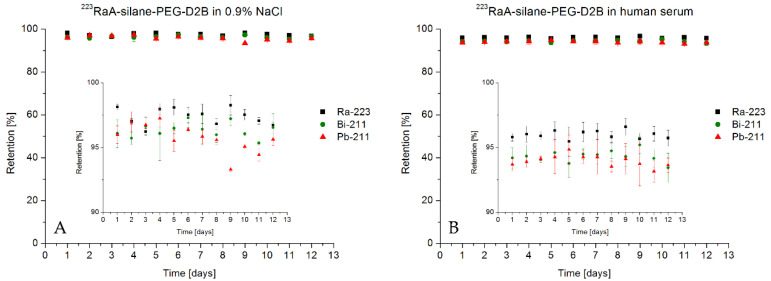
Retention of ^223^Ra, ^211^Bi, and ^211^Pb on NaA-silane-PEG-D2B product radiolabeled with ^223^Ra in (**A**) 0.9% NaCl and (**B**) and human blood. Insets show the values of retention from 90% to 100%. Stability was determined in 100 µL of ^223^RaA-silane-PEG-D2B, which represented 100 µg/mL and 0.065 MBq.

**Figure 7 materials-13-03875-f007:**
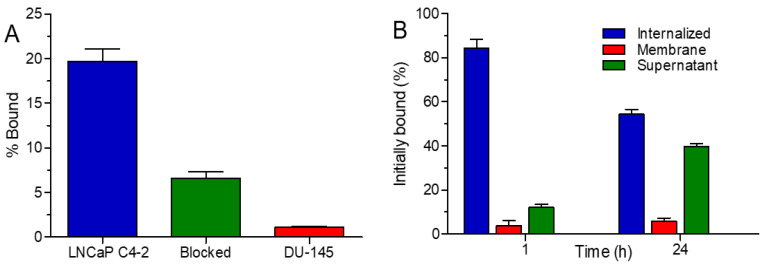
(**A**) The binding specificity of ^131^I-D2B-PEG-silane-NaA to LNCaP C4-2 and DU145 cells without and in the presence of 25-fold molar excess of unlabeled D2B mAb (Blocked); (**B**) Different cellular fractions of ^131^I-D2B-PEG-silane-NaA during receptor-mediated internalization into LNCaP C4-2 cells over 24 h at 37 °C.

**Figure 8 materials-13-03875-f008:**
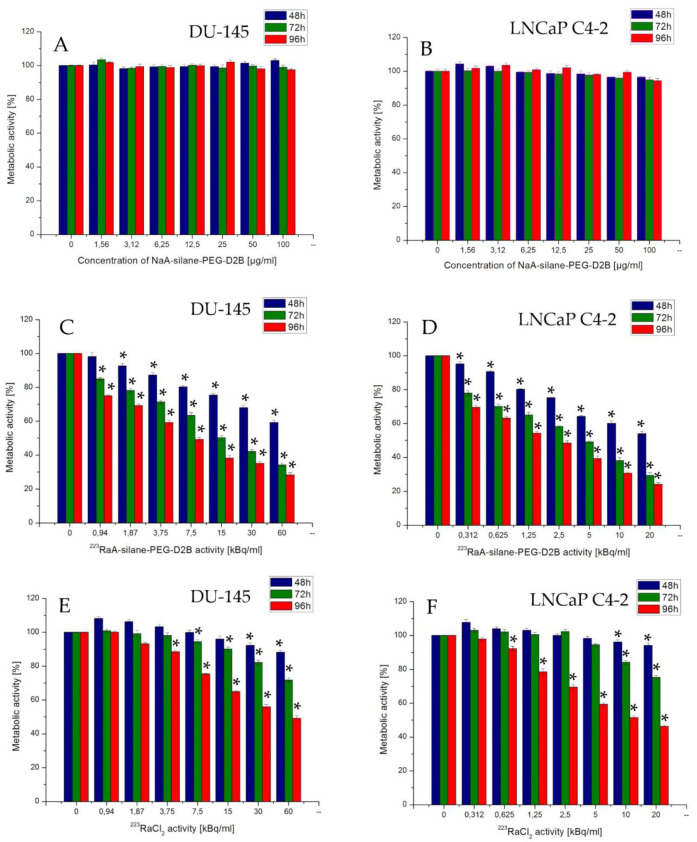
Metabolic activity of DU-145 and LNCaP C4-2 cells cells treated with different concentrations of immunoconjugate NaA-silane-PEG-D2B (**A**,**B**), radioimmunoconjugate ^223^RaA-silane-PEG-D2B (**C**,**D**) and ^223^RaCl_2_ (**E**,**F**) for 48 h, 72 h, or 96 h. Data are expressed as a percent of control, mean  ±  SD from three independent experiments (*** denotes statistically significant difference from unexposed control, *p* < 0.05).

**Table 1 materials-13-03875-t001:** The gel compositions and conditions of NaA nanozeolite synthesis. (* aging process at room temperature, # aging process at a temperature of an ice bath).

Sample	Gel Composition	Reaction Time (h)	Crystallization Temperature (°C)	Crystallization Time (h)
1	1.11 Na_2_O:0.16 Al_2_O_3_:1 SiO_2_:85.97 H_2_O: 2.21 TMAOH	48 *	100	24
2		96 *	100	5
3		48 *	40	24
4		48 *	60	24
5	0.16 Na_2_O:1 Al_2_O_3_:6 SiO_2_:350 H_2_O: 14.56 TMAOH	96 *	100	24
6		96 #	100	24

**Table 2 materials-13-03875-t002:** Characteristics of NaA samples: average crystal size according to SEM data and crystallinity according to X-ray diffraction.

Sample	Average Crystal Size (nm)	Crystal Size Distribution (nm)	Crystallinity (%)
1	651.40 ± 163.24	600–800	80
2	549.80 ± 132.03	500–700	65
3	88.72 ± 46.12	30–100	Amorphous
4	130.33 ± 44.75	50–150	30
5	240.63 ± 92.52	50–300	94
6	120.24 ± 28.33	50–150	95

**Table 3 materials-13-03875-t003:** The effect of time and temperature of calcination on porous parameters of NaA nanozeolite (sample 6). S_BET_-surface area, V_t_-total (single-point) pore volume obtained from the amount adsorbed at p/po = 0.99, D_ave_-average pore diameter, V_micro_-micropore volume (pores < 2 nm), and microporosity.

Time [h]	Temperature [°C]	S_BET_ [m^2/^g]	D_ave_ [nm]	V_t_ [cm^3^/g]	V_micro_ [cm^3^/g]	Microporosity [%]
3	500	428.52	2.58	0.27	0.179	64.80
72	600	485.03	2.88	0.35	0.217	62.14

**Table 4 materials-13-03875-t004:** Hydrodynamic diameter, zeta potential (ζ), and polydispersity index of the obtained samples (measured in PBS pH = 7.4). Hydrodynamic diameter is determined by the NTA method in RPMI 1640 medium (*) and in EMEM medium (#).

Sample	Hydrodynamic Diameter [nm]	Zeta (ζ) Potential [mV]	Polydispersity Index
NaA	196.7 ± 53.5 * 206.0 ± 40.3 #	−40.3 ± 2.11	0.292 ± 1.21
NaA-silane PEG	216.9 ± 56.62 * 217.1 ± 34.22 #	−43.1 ± 2.45	0.202 ± 1.01
NaA-silane-PEG-D2B	225.1 ± 48.61 * 226.1 ± 44.21 #	−45.08 ± 1.97	0.186 ± 1.04

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
