# Peer review of "Design and Evaluation of 223Ra-Labeled and Anti-PSMA Targeted NaA Nanozeolites for Prostate Cancer Therapy–Part I"

_materials, 2020, doi:10.3390/ma13173875_

Round 1
Reviewer 1 Report
Dear authors, the proposed work seems well written and structured.
SEM images should be improved in quality and probably scale bar should be revised.
Some images related to a single particle should be also interesting, since the obtained images are quite like aggregates.
I would suggest removing "novel" from the title, given the growing literature on the subject.
Reviewer 2 Report
In this manuscript the authors have designed 223RaA-silane-PEG-D2B and evaluated its efficacy in prostate cancer cells. The study is well designed and has reproducible results.
It would be very intresting to know from authors in the discussion the possible way 223RaA-silane-PEG-D2B could be made efficacious in the DU-145 cells.
Also The authors may need to show if 223RaA-silane-PEG-D2B has any effect on normal cells.

Reviewer 3 Report
This is indeed a very interesting paper, but also very complicated. The overall aim at efficient, alpha particle mediated internal radionclide therapy of prostate cancer is of course of outmost importance. The indentification of Ra-223 as a solution that bypasses the supply bottleneck of other therapeutic alpha emitters is brave.
At first sight, I did not believe it would be possible to contain the entire decay chain of Ra-223 inside these therapeutic nano-particles, with labile Rn-219 just one step down the chain. But the particle design, and the data presented has convinced me.
I can see that this manuscript is only Part I of a sequel, and I may ask for revisions that are actually delivered/clarified in a part II ( that I have not seen).
However, if the editor or another reviewer asks for a major revision, I think this part I could benefit from addition of some dosimetry, ... that is: estimates of the absorbed dose calculated for the kind activitiy concentration, you use in the cell experiments. Are the radiobiological effects claimed plausible with such absorbed doses?
This said, I have some important points that I think you must address in a minor revision:
A) Are these funtionalised nano-particles small enough to reach the biological target and penetrate sufficiently the targeted tumour cells?
B) Are the particles intended for systemic therapy ? In case, is it likely that they can circulate sufficiently long to reach the necessary specific accumulation ? What other tissues can be expected to get a severe hit by particle accumulation/clearance?
C) Some feasability must be added in terms of the number of Ra-223 decays/chain alphas to "kill" a targeted cell ?
D) Internalisation is described as a likely event for these PSMA directed entities, but is internalisation necessary at all to achieve the therapeutic endpoint? Alphas do have ranges in the order of several cell diameters.
E) The figure 6 is to me the most important and convincing. But can you clarify how well the the applied gammacounter can discriminate the various isotopes, and what gamma lines / intensities you are using.
F) And finally: Why can you avoid the Rn-219 escape from such small particles ? It may be that it is actually a result of re-uptake of Rn-219 and daughters into (other) nano particles. In such case, the Rn-219 mediated activity spread will be much much worse in any realistic in-vivo case, with low particle concentrations, perfusion, etc. If the reader is to judge this, you must supply data on both the particle concentration and the activity concentration used in the stability studies in fig.6. Was agitation used, are the particles suspended during the entire stability period?
And some minor, gritty , details:
0) The Title, ... the word "nanozeolite" indicates something in the singular sense, but as it is indeed many particles, used, perhaps the title should be changed to the plural sense?
1) Line 23: It is not the therapies that "lead" to recurrence, But inadeqacy of the therapies.
2) line72-73: you writhe "... suboptimal for eradication of single metastatic cells or small cell clusters...." ...Well this is a hypothesis we all like to believe, but is it true. Can you substantiate the hope with a refernce ?
3) Likewise, in line 80:"....or individual, circulating neoplastic cells...." Again a hypothesis deserving some reference.
4) line 107-108, Here you must be fair and add, that it is not only the lack of chelators limiting the use of Ra-223, but also very real concerns about the but also the Rn-219 escape from the targeted region.
5) Line 180: What is you supply source source of the Ac-227 for this study ?
6) Line 246: Clarify what you mean by "real" parameters?
7) line 192..The 223Ra labeled nanozeolites were washed several times in distilled water to remove unbounded 223-Ra isotope [33]. Is this not an eternal proces, where you can keep washing out Ra-223 given sufficient amounts of water? Or is the zeolote binding of the Ra-223 really "irreversible?
8)Preparation of the "antibody" ..line 220---232.. Can you give a bit more details of the antibody ? mw for instance?, stability? isoelectric point?
9) Lines 236--247, ...I am not an expert on electron microscopy, just aplain nuclear physicist, but to me there seem to be many different high voltages. What are they all, used for? Can the section be simplified?
10) Line 245... How do you get a ( calibrated)(average) particle size from SEM images?
11) Line 305 .. Please state WHAT radionclides you quantify.
12) Line 561..".was measured on γ-spectrometer". iS THIS THE AUTOMATIC SAMPLE CHANGER GAMMA COUNTER mentioned later ? Is it a scintillation counter (type?) or ?
13) figure 6 A and B ... Please give concentration of particles in the incubation media.
14) line 569 f : Do the nanoparticles really internalize? Perhaps you have SEM images showing that?
But even if particles internalise to a very high dergree, how can you argue that the single cell wall will help contain Rn-219?
15) The I-131 2control/competition/ localisation experiments are really nice. But can you add some convincing words that the I-131 does not change the "function" and "distribution" of the D2B antibody ?
16) Line 661f: As said before:... internalisation not necessarily a benefit for alpha therapy.
17) line 692: Patents: Section is blank. Do you mean that none of the described work is covered by any patents?
18) line 806: incomplete refernce :Westrøm et al., 2018.pdf.
But in total: Very thorough, brave, interesting work !
